# Method of Calculating Inductance Gradient for Complex Electromagnetic Rail Launcher

**Shida Ren** [1], **Gang Feng** [2,*]**, Pengxiang Zhang** [1], **Tengda Li** [1] **and Xilai Zhao** [1]

1   Graduate College, Air Force Engineering University, Xi'an 710051, China
2   Air and Defense College, Air Force Engineering University, Xi'an 710051, China
*   Correspondence: yuanshangren001@163.com; Tel.: +86-133-8929-2021

**Abstract:** Electromagnetic rail launch technology has made impressive progress; however, the analytical method of calculating the inductance gradient for a complex electromagnetic launcher is still insufficient. By fully considering the characteristics of electromagnetics and current distribution in a device, this paper describes a model of the current skin effect by simplifying the line current distribution in the device. Based on Biot–Savart Law, an analytical method of calculating inductance gradient for an electromagnetic rail launcher with complex structure is proposed. This method has the advantages of fast calculation speed and accurate calculation results. Because of error analysis, the calculated value relatively corresponds to the simulation result of the eddy current field. To reflect the transient electromagnetic emission process, the effects of different configurations, current frequency, and armature position on the inductance gradient are further summarized. The results show that the error rate of this method in calculating the inductance gradient is about 4%, which meets the requirement for calculation accuracy. The inductance gradient of the enhanced four-rail electromagnetic launcher is about 2.22 times that of the nonenhanced one due to the equal conditions; the inductance gradient decreases with the increase in current frequency and decreases as the armature approaches the muzzle.

**Keywords:** electromagnetic rail launcher; Biot–Savart Law; current skin effect; inductance gradient

## 1. Introduction

Electromagnetic launch is a newly developed weapon launch technology that uses electromagnetic force to accelerate the projectile to ultrahigh sound velocity [1–5]. This technology can break through limitations on the speed of traditional gunpowder and it accurately controls the muzzle velocity of a projectile by controlling the pulse current [6,7].

The inductance gradient is an important parameter in the design of electromagnetic rail launchers. This parameter is directly related to the size of the electromagnetic thrust, which affects the muzzle velocity of the projectile and restricts the setting of launcher structure and power supply parameters. Currently, there are two inductance gradient calculation methods. One is the analytical calculation method based on Biot–Savart Law, and the other is the finite element simulation solution method. In the structural parameter optimization process of electromagnetic rail launchers, it is necessary to calculate the inductance gradient of numerous models. Because the finite element simulation method has requires extensive calculation resources and a long calculation time, it is not often used. In contrast, the analytical method can quickly achieve this purpose. Therefore, research on the analytical calculation method of the inductance gradient is of great significance [8].

A large volume of research has been conducted on the numerical analytical method of calculating the rail inductance gradient. Kerrisk [9] calculated the current distribution in the rectangular guide rail of an electromagnetic launcher and proposed an analytical method for calculating its high-frequency inductance gradient. The calculation formula is given as

$$L' = [(0.4406 - 0.777 \ln(F_1)) \ln(F_2)],$$

where

$$\begin{cases} F1 = 3.397(wh) - 0.06603(wh)(sh) + 1 \\ F2 = 2.7437(sh) + 0.02209(wh) + 0.2637(wh)(sh) + 1.0077. \end{cases}$$

Additionally, Grover [10] proposed a simple method to calculate the low-frequency inductance gradient of a rectangular guide rail. The formula is as follows:

$$L' = 0.4 \left[ \ln(\frac{s+w}{h+w}) + \ln k + 1.5 \right].$$

These two methods are theoretical derivations calculated on the premise of regular rail geometry. The inductance gradient is calculated under ideal conditions. In the actual launch process, the strong pulse current makes the process extremely complex. The inductance gradient of a guide rail is affected by factors such as eddy current loss, armature position, and current frequency.

For the above two limitations, various scholars have conducted targeted research. Wu et al. [11] considered the velocity skin effect and assumed that the current was evenly distributed on the inner surface and upper and lower surfaces of the rail, and they deduced the formula for calculating the inductance gradient of a simple structure two-rail electromagnetic launcher. However, their calculation method did not reflect the influence of current skin depth on the inductance gradient. Kim et al. [12] studied the influence of the size of armature–rail structures on the inductance gradient through the finite element method. Xu et al. [13] calculated the inductance gradient of an enhanced electromagnetic railgun using finite element analysis software, and they summarized the law of the influence of the geometry of the enhanced electromagnetic railgun on the inductance gradient. Peng et al. [14] established an equivalent geometric model based on the skin depth, and they simulated the static magnetic field of the model to solve the inductance gradient, but this method is not suitable for complex armature–rail structures.

In this study, through finite element simulation, the electromagnetic and current distribution characteristics of an electromagnetic launcher with a complex structure were researched. Based on Biot–Savart Law, a simplified model of the current skin effect was deduced from the inductance gradient analytical method. For the two configurations of the electromagnetic launcher, the inductance gradient obtained using eddy current field simulation was used as a comparison for error analysis. The effects of different configurations, current frequency, and the armature position on inductance gradient were further summarized. These effects provide a theoretical basis for calculating the transient inductance gradient of electromagnetic rail launchers with complex structures.

## 2. Electromagnetic Characteristic Analysis

To make the analytical method more truly reflect the actual emission situation, it was necessary to comprehensively consider the current and electromagnetic distribution in the device when calculating the inductance gradient. Therefore, the current distribution characteristics and electromagnetic environment of the complex four-rail electromagnetic launcher were simulated and analyzed.

### 2.1. Simulation Model and Simulation Conditions

The 3D models of enhanced and nonenhanced four-rail electromagnetic launchers used in this study are shown in Figure 1. The enhanced four-rail electromagnetic launcher correspondingly had an auxiliary rail outside the nonenhanced four rails. The whole electromagnetic transmitter used only one set of power supply devices. The adjacent rail had the opposite current direction, whereas the auxiliary rail had the same current direction as its corresponding main rail. The main rail carried the armature movement, whereas the auxiliary rail enhanced the magnetic field in the gun bore. Compared with the nonenhanced four-rail electromagnetic launcher, the enhanced type can obtain higher muzzle velocity under the same pulse current intensity.

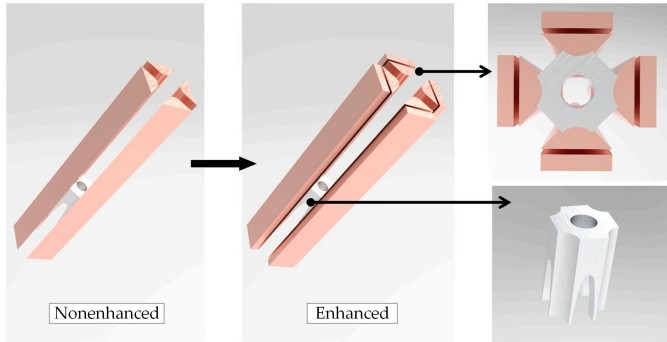

**Figure 1.** The 3D structure diagram.

The eddy current field simulation solver can calculate the current frequency in the range of zero to hundreds of megahertz. Furthermore, the solver can better reflect the influence of current skin and proximity effects on inductance gradient in the emission process. The excitation current frequency was 1 kHz, and the amplitude was 400 kA. The material parameters are given in Table 1.

**Table 1.** Material parameter setting.

| Material | Density $(\mathbf{kg \cdot m^{-3}})$ | Vacuum Permeability $(\mathbf{H \cdot m^{-1}})$ | Conductivity $(\mathbf{S \cdot m^{-1}})$ | Resistivity $(\mathbf{kg \cdot m^{-3}})$ |
|---|---|---|---|---|
| Copper | $8.66 \times 10^3$ | $1.25664 \times 10^{-6}$ | $5.8 \times 10^7$ | $5.4 \times 10^{-8}$ |
| Aluminum | $2.6989 \times 10^3$ | $1.25664 \times 10^{-6}$ | $3.76 \times 10^7$ | $2.6548 \times 10^{-8}$ |

*2.2. Simulation Results and Analysis*

Current density distribution is an important electromagnetic characteristic of electromagnetic rail launchers. The current density distribution of armature–rail structures affects the distribution of the internal magnetic field intensity of an electromagnetic rail, and it reflects the distribution of the force and heat source of the launcher. Additionally, it has an important influence on the calculation of the inductance gradient. Because the current density distribution of the four-rail electromagnetic launcher has central symmetry, the quarter rail was analyzed, and the current density cloud diagram is shown in Figure 2.

Figure 2 shows that both types of devices reflected the current skin and proximity effects to varying degrees. The current in the rail of a nonenhanced four-rail electromagnetic launcher was mainly concentrated on the outside surface of the rail, and the current density in the central area became insignificant and almost zero. From the cross-section, the current density presented an O-shape distribution.

The distribution of rail current density changed after adding a layer of auxiliary rail outside the non-enhanced four-rail electromagnetic launcher. Owing to the influence of the proximity effect when a pair of main and auxiliary rails is connected with the current in the same direction, the current density of the main rail concentrated on the side close to the armature, whereas the current density of the auxiliary rail concentrated on the side away from the armature. From the cross-section, the overall current density of the main and auxiliary rails presented an O-shape distribution.

The current density in the armature was mainly distributed near the current excitation end, i.e., the current skin layer. The maximum current density was distributed at the inner throat of the armature, and the current density at the armature head was almost zero, which was the result of the current conduction along the shortest path, and the armature current distribution of the two structures was consistent. The armature current density distribution and the current density distribution on the contact surface of the armature rail are shown in Figure 3.

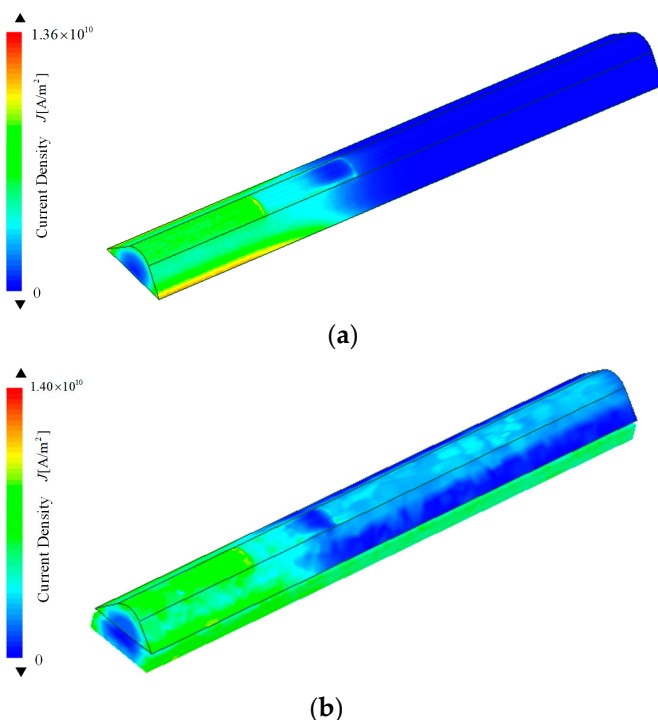

**Figure 2.** Current density distribution: (**a**) nonenhanced rail; (**b**) enhanced rail.

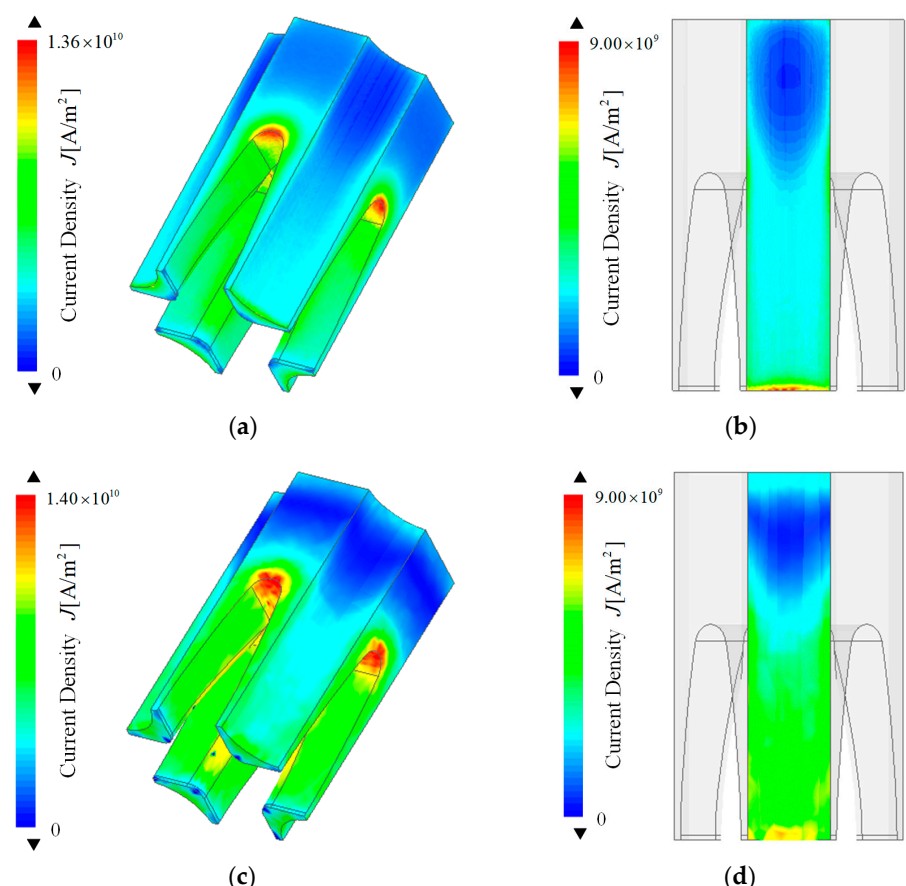

**Figure 3.** Armature and contact surface current density distribution: (**a**) nonenhanced armature; (**b**) nonenhanced armature and rail contact surface; (**c**) enhanced armature; and (**d**) enhanced armature and rail contact surface.

The current density in the armature was mainly distributed near the current excitation end, that is, the current skin layer. The maximum value of current density was distributed at the inner throat of the armature, and the current density of the armature head was almost zero, which was the result of the current conducting along the shortest path, and the armature current distribution of the two structures was consistent. On the contact surface of the armature rail, the current density at the tail of the armature arm concentrated, the nonenhanced current density concentration phenomenon was more obvious, and the enhanced current density distribution was more uniform. Affected by the current density distribution of the secondary rail, the current density on the enhanced armature arm was greater than that of the nonenhanced armature arm. Based on the above analysis, the current density distribution on the armature of the enhanced four-rail electromagnetic transmitter was better than that of the nonenhanced one.

To more clearly see the current skin depth, we selected the axial section inside the transmitter. The axial current distribution is shown in Figure 4.

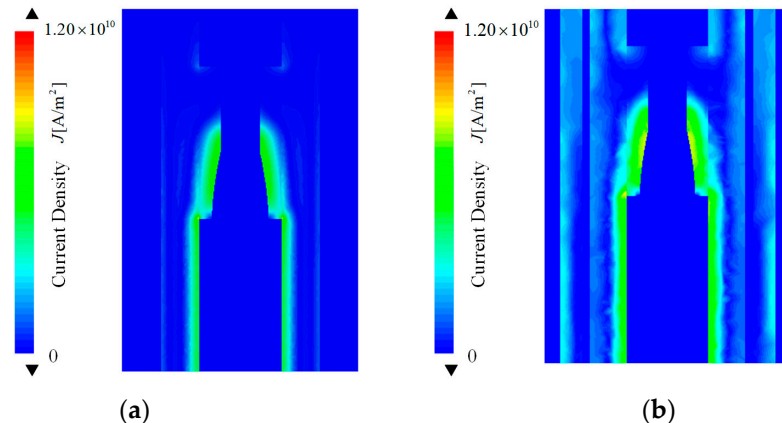

(a)  (b)

**Figure 4.** Axial section of magnetic field intensity distribution: (**a**) nonenhanced; (**b**) enhanced.

According to the cross-sectional current density nephogram, the skin effect made the current evenly distributed on the thin layer area on the inner surface of the armature and rails, and the skin thickness of the armature was greater than that of the rails, which agreed with the calculation law of the skin depth formula.

According to the above analysis, analysis of the electromagnetic characteristics in the bore was essential. To more intuitively see the armature magnetic field distribution, we selected the front and rear-end faces of the armature, as shown in Figure 5, and simulated the cloud diagram of its magnetic field intensity distribution.

Figure 5 shows that the magnetic field intensity distribution of the armature end face of the two devices was similar and concentrated on the contact surface between the armature and the rail. There was always current conduction in the auxiliary rail of the enhanced four-rail electromagnetic launcher so that the magnetic field intensity of the enhanced type was significantly higher than that of the nonenhanced type. A circular magnetic field shielding area formed in the central area of the bore, which can effectively meet the requirements of intelligent weapons and ammunition in electromagnetic environments.

To visually reflect changes in the magnetic field intensity in the bore at different positions, we took the symmetry axis of the front and rear-end faces of the armature as the path and analyzed the magnetic field intensity distribution on the path. The magnetic field intensity distribution curve on each path is shown in Figure 6.

Figure 6 shows that the magnetic induction intensity of the two structures was symmetrically distributed in the path. The magnetic induction intensity at the rear end of the armature of the electromagnetic launcher was much greater than that at the front end of the armature, which was caused by the principle of the shortest current so that the electromagnetic intensity was mainly concentrated in the area below the armature. For the interior of the gun chamber of the enhanced four-track electromagnetic launcher, the

magnetic induction intensity within the range of path [18 mm, 22 mm] was 0.99~19.21 T; the magnetic induction intensity in the range of the nonenhanced path [10 mm, 32 mm] was 0~14.49 T. The maximum value of the bottom end of the nonreinforced armature was about seven times that of the top end, while the ratio of the reinforced armature was only four times larger, which was the superposition effect of the current in the auxiliary rail on the magnetic field in the bore. Therefore, under the same parameters, the enhanced four-rail electromagnetic launcher can strengthen the magnetic field intensity in the bore, and the magnetic field intensity distribution is more uniform, which can effectively improve the electromagnetic emission efficiency.

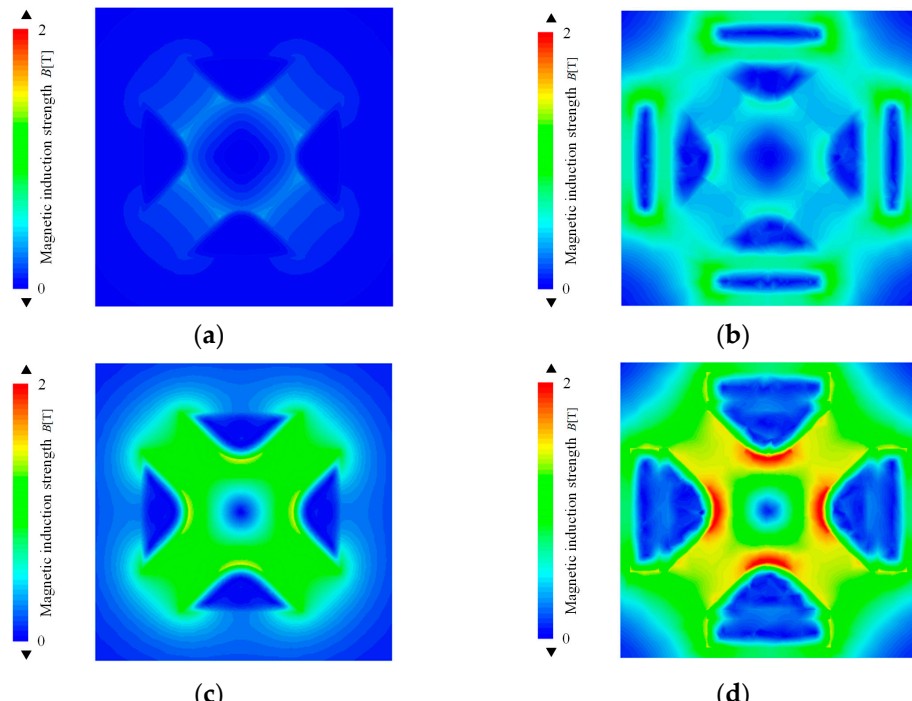

**Figure 5.** Magnetic field intensity distribution of the radial section: (**a**) nonenhanced front-end face; (**b**) enhanced front-end face; (**c**) nonenhanced rear-end face; (**d**) enhanced rear-end face.

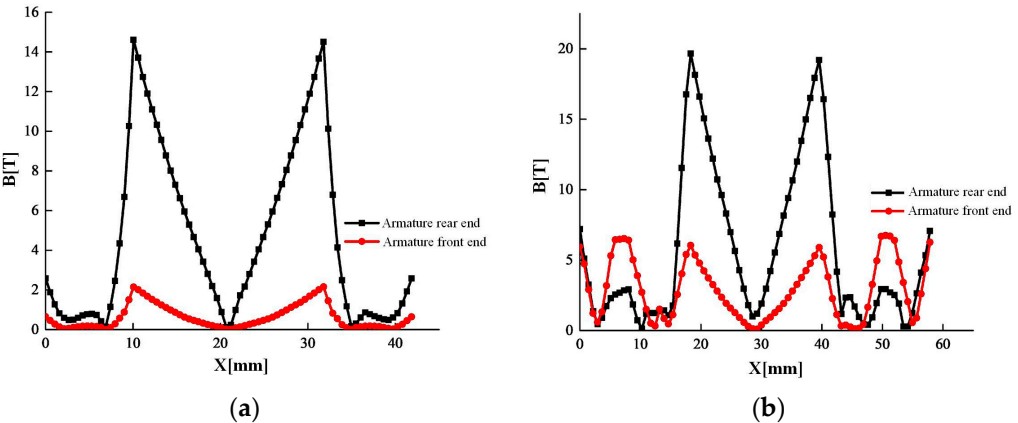

**Figure 6.** Distribution curve of magnetic induction intensity on the path: (**a**) nonenhanced; (**b**) enhanced.

## 3. Derivation of Analytical Formula

### 3.1. Model Simplification

In the actual working process of an electromagnetic rail launcher, the energization time is very short, and the current is affected by the skin effect so that it cannot be evenly distributed in the armature and rail. From the current density distribution, the current

is concentrated on the thin layer area inside the armature and rails. According to the inductance gradient formula, only the current that excites the electromagnetic force in the +Z direction impacts the inductance gradient, while the current in the armature arm is mainly used to generate lateral electromagnetic force. Therefore, while reflecting the actual launch situation, the theoretical derivation process was simplified, and only the skin layer of the armature head was modeled. The simplified model of armature axial direction is shown in Figure 7a.

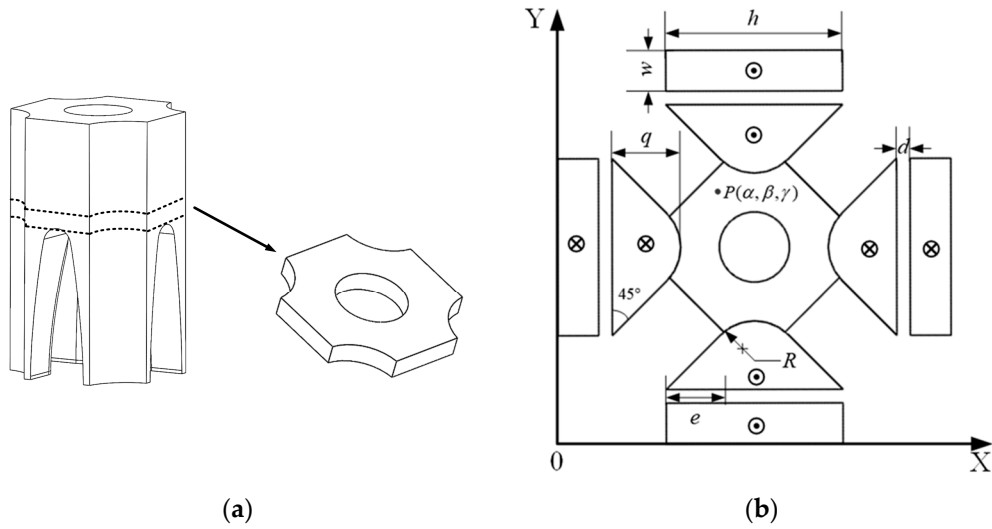

(**a**)                (**b**)

**Figure 7.** Simplified model of skin effect. (**a**) Simplified model of armature axial direction; (**b**) schematic of parameter annotation.

### 3.2. Derivation of Nonenhanced Inductance Gradient Formula

Next, starting with Biot–Savart Law, the inductance gradient analytical formula of the model was deduced. The schematic of parameter annotation is shown in Figure 7b.

The pulse current *I* is introduced into the rail, and the current in the rails generates a quadrupole magnetic field in the bore. The magnetic field interacts with the current flowing through the armature to produce electromagnetic thrust in the +Z direction, i.e., the inward direction perpendicular to the X–Y plane shown in Figure 7. Considering the skin effect, we assumed that the current is evenly distributed in the skin layer of the guide rail. The cross-sectional area of the skin layer of the main rail is $S_1$ and that of the skin layer of the auxiliary rail is $S_2$. The skin layer of current is a uniformly distributed line current. According to Biot–Savart Law,

$$d\boldsymbol{B} = \frac{\mu_0}{4\pi} \frac{I d\boldsymbol{l} \times \boldsymbol{r}}{|\boldsymbol{r}|^3} \tag{1}$$

where *l* is the length of the energized conductor, $\mu_0$ is the vacuum permeability, $I d\boldsymbol{l}$ is the current element, $\boldsymbol{r}$ is the vector from $I d\boldsymbol{l}$ to *P*, and $d\boldsymbol{B}$ is the magnetic induction intensity generated by $I d\boldsymbol{l}$ at point *P*.

The electromagnetic analysis of main rail 1 is conducted below. According to the known rail parameters, the algebraic relationship of the upper surface of rail 1 cross-section is established as follows:

$$f_1(x) = \begin{cases} x + w + d - \dfrac{D-h}{2}, \dfrac{D-h}{2} \le x < e + \dfrac{D-h}{2} \\ \sqrt{R^2 - \left(x - \dfrac{D}{2}\right)^2} + q - R + w + d, e + \dfrac{D-h}{2} \le x < \dfrac{D+h}{2} - e \\ -x + w + d + \dfrac{D+h}{2}, \dfrac{D+h}{2} - e \le x \le \dfrac{D+h}{2} \end{cases} \tag{2}$$

where $D$ is the muzzle diameter of the enhanced four-rail electromagnetic launcher. Because the current in the rail flows along the z-axis direction, the current element at any point on the main rail 1 can be expressed as $(I/S_1)dxdy\mathbf{k}$, where $\mathbf{i}, \mathbf{j}, \mathbf{k}$ represent the unit vectors on the three-coordinate axes, and $S_1$ can be obtained from the skin depth formula. $I$ is the effective value of the current. Taking any point $P(\alpha, \beta, \gamma)$ on the armature, the magnetic field strength of this point is given as

$$d\mathbf{B}_1 = \frac{\mu_0}{4\pi} \frac{I dxdy\mathbf{k} \times \mathbf{r}}{S_1|\mathbf{r}|^3} \tag{3}$$

where $\mu_0$ is the vacuum permeability. Because the selected current element tends to be infinitesimal, the vector from the current element to point $P$ can be expressed as

$$\mathbf{r} = (x - \alpha)\mathbf{i} + (y - \beta)\mathbf{j} + (z - \gamma)\mathbf{k} \tag{4}$$

$(x, y, z)$ is the coordinate of the center point of the current element, and the mode length of the vector is given as

$$|\mathbf{r}| = \sqrt{(x - \alpha)^2 + (y - \beta)^2 + (z - \gamma)^2} \tag{5}$$

Then, Formula (3) can be further expressed as

$$d\mathbf{B}_1 = \frac{\mu_0}{4\pi} \frac{I dxdy}{S_1|\mathbf{r}|^3}[(x - \alpha)\mathbf{k} \times \mathbf{i} + (y - \beta)\mathbf{k} \times \mathbf{j}] \tag{6}$$

By integrating Equation (6), we obtained that the magnetic field intensity generated by the current in the main rail 1 at point $P$ is $\mathbf{B}_1$. The magnetic field intensity generated by any guide rail at point $P$ is $\mathbf{B}_i$. According to the vector superposition principle of a magnetic field, we concluded that the magnetic field intensity generated by the current in the guide rail at the point $P$ is $\mathbf{B}$.

$$\mathbf{B}_1 = \frac{\mu_0 I}{4\pi} \frac{1}{S_1\left[(x - \alpha)^2 + (y - \beta)^2 + (z - \gamma)^2\right]^{\frac{3}{2}}} \cdot \int_{\frac{D-h}{2}}^{\frac{D+h}{2}} \int_{w+d}^{f(x)} [(x - \alpha)\mathbf{k} \times \mathbf{i} + (y - \beta)\mathbf{k} \times \mathbf{j}]dxdy \tag{7}$$

$$\mathbf{B}_i = \frac{\mu_0 I}{4\pi} \frac{(-1)^{i+1}}{S_1\left[(x - \alpha)^2 + (y - \beta)^2 + (z - \gamma)^2\right]^{\frac{3}{2}}} \cdot \int_{\frac{D-h}{2}}^{\frac{D+h}{2}} \int_{w+d}^{f(x)} [(x - \alpha)\mathbf{k} \times \mathbf{i} + (y - \beta)\mathbf{k} \times \mathbf{j}]dxdy \tag{8}$$

$$\mathbf{B} = \frac{\mu_0 I}{4\pi} \sum_{i=1}^{4} \frac{(-1)^{i+1}}{S_1\left[(x - \alpha)^2 + (y - \beta)^2 + (z - \gamma)^2\right]^{\frac{3}{2}}} \cdot \int_{\frac{D-h}{2}}^{\frac{D+h}{2}} \int_{w+d}^{f(x)} [(x - \alpha)\mathbf{k} \times \mathbf{i} + (y - \beta)\mathbf{k} \times \mathbf{j}]dxdy \tag{9}$$

According to the calculation formula of magnetic field intensity in the bore, the force on the current element on the armature can be integrated to obtain the electromagnetic force on the armature, i.e.,

$$\mathbf{F}_{1i} = \int_{l_i} I\mathbf{B}_i \times d\mathbf{l} \tag{10}$$

According to the armature current vector distribution, the current density around the armature guide hole is low. Therefore, to deduce the electromagnetic thrust more accurately and simplify the calculation, the circular ammunition loading area was simplified to a square area. The simplified armature and current flow are shown in Figure 8.

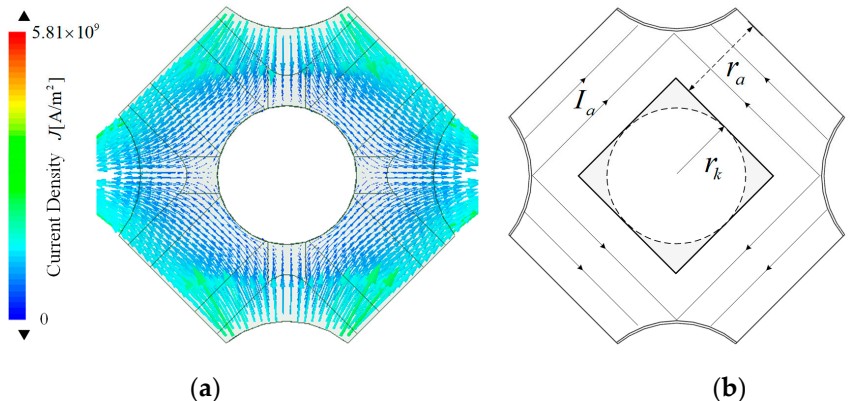

**Figure 8.** Simplified process of armature current. (**a**) Current distribution; (**b**) simplified current model.

According to the current vector distribution in the armature, the excitation current is symmetrically distributed in the armature, so the current in every quarter of the armature is $I_a = I/2$, and the current density flowing through the armature section is $J = I_a/(r_a \cdot a)$, where $a$ is the axial skin depth of the armature head, and the skin depth calculation formula is given as

$$a = \sqrt{\frac{1}{\pi f \mu \sigma}} \qquad (11)$$

where $f$ is the current frequency, $\mu$ is the vacuum permeability, and $\sigma$ is the conductivity. For example, when the current frequency is 1 kHz, the calculated skin depths of aluminum and copper are 2.6 and 2.1 mm, respectively.

The electromagnetic thrust acting on the armature is the vector sum of the electromagnetic thrust generated by the four rails on the armature. Owing to the symmetry of the armature–rail structure and the current being the same in each rail, the electromagnetic force of each rail on the armature is the same. Then, the electromagnetic thrust received by the armature is given as

$$
\begin{aligned}
F_1 &= \int_{l_i} \sum_{i=1}^{4} B_i J \times dl \\
&= \int_{l_i} \frac{\mu_0 I J}{4\pi} \sum_{i=1}^{4} \frac{(-1)^{i+1}}{S_1 \left[(x-\alpha)^2+(y-\beta)^2+(z-\gamma)^2\right]^{\frac{3}{2}}} \cdot \int_{\frac{D-h}{2}}^{\frac{D+h}{2}} \int_{w+d}^{f(x)} [(x-\alpha)k \times i + (y-\beta)k \times j] dx dy \times dl
\end{aligned} \qquad (12)
$$

The inductance gradient of the nonenhanced electromagnetic launcher is given as

$$L' = \frac{2F_1}{I^2} \qquad (13)$$

### 3.3. Derivation of Enhanced Inductance Gradient Formula

The device has an additional layer of auxiliary rails based on a nonenhanced four-rail electromagnetic launcher. When calculating the inductance gradient of the enhanced launcher, the magnetic field generated by the four subrails is coupled and superimposed with the nonenhanced launcher, and the current distribution through the armature remains unchanged.

The magnetic induction intensity formula of the four auxiliary rails to any point in the gun bore is given as

$$
\begin{aligned}
B' &= \sum_{i=1}^{4} B_i' \\
&= \frac{\mu_0 I}{4\pi} \sum_{i=1}^{4} \frac{(-1)^{i+1}}{S_2 \left[(x-\alpha)^2+(y-\beta)^2+(z-\gamma)^2\right]^{\frac{3}{2}}} \cdot \int_{\frac{D-h}{2}}^{\frac{D+h}{2}} \int_{0}^{w} [(x-\alpha)k \times i + (y-\beta)k \times j] dx dy
\end{aligned} \qquad (14)
$$

The electromagnetic thrust provided by the armature by the auxiliary rail is given as

$$
\begin{aligned}
F_2 &= \int_{l_i} \mathbf{B}' J \times d\mathbf{l} \\
&= \int_{l_i} \frac{\mu_0 I J}{4\pi} \sum_{i=1}^{4} \frac{(-1)^{i+1}}{S_2 \left[ (x-\alpha)^2 + (y-\beta)^2 + (z-\gamma)^2 \right]^{\frac{3}{2}}} \cdot \int_{\frac{D-h}{2}}^{\frac{D+h}{2}} \int_0^w \left[ (x-\alpha)\mathbf{k} \times \mathbf{i} + (y-\beta)\mathbf{k} \times \mathbf{j} \right] dx\,dy \times d\mathbf{l}
\end{aligned}
\tag{15}
$$

The electromagnetic thrust of the enhanced armature is given as

$$
F_z = F_1 + F_2 \tag{16}
$$

The inductance gradient of enhanced electromagnetic launcher is given as

$$
L'' = \frac{2F_Z}{I^2} \tag{17}
$$

## 4. Simulation Calculation Method of Eddy Current Field

Many studies show that the inductance gradient obtained using the eddy current field simulation method is close to that obtained from the experimental results. Therefore, in this section, we compare the simulation results of the eddy current field to verify the accuracy of the analytical calculation method.

*Comparative Analysis of Two Calculation Methods*

To further verify the accuracy of the analytical method, the two abovementioned methods were used to calculate the inductance gradient for a specific model. The structural parameters of the selected example model are shown in Table 2.

**Table 2.** Structural parameters of calculation example.

| $D(\mathrm{mm})$ | $h(\mathrm{mm})$ | $w(\mathrm{mm})$ | $d(\mathrm{mm})$ | $q(\mathrm{mm})$ |
|---|---|---|---|---|
| 57.86 | 26 | 6 | 2 | 10.07 |
| $R(\mathrm{mm})$ | $e(\mathrm{mm})$ | $L_{rail}(\mathrm{mm})$ | $L_{Armature}(\mathrm{mm})$ | $r_a(\mathrm{mm})$ |
| 7.07 | 8 | 217.3 | 40 | 6.855 |
| $r_k(\mathrm{mm})$ | $f(\mathrm{Hz})$ | $I(\mathrm{kA})$ | - | - |
| 5.175 | 1000 | 400 | - | - |

The eddy current field simulation method was used to solve the energy and electromagnetic thrust of the enhanced type at different positions of the armature. The solution results of the enhanced type are shown in Figure 9.

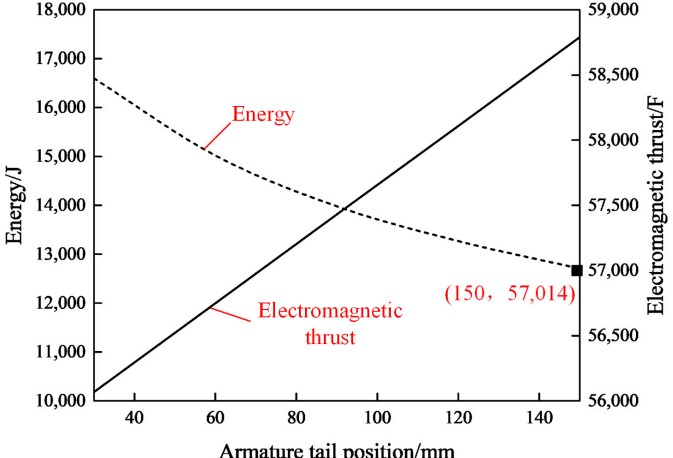

**Figure 9.** Simulation results of enhanced eddy current field.

We selected the armature head at 100 mm and used the current excitation with a frequency of 1000 Hz and peak value of 400 kA to solve the electromagnetic force and inductance gradient, respectively. The error between the results and the analytical method is shown in Table 3. The error rate was obtained by dividing the difference between the simulation and analytical values by the simulated value.

**Table 3.** Comparison of calculation results of two methods.

| Parameter | Configuration | Analytical Value | Simulation Value | Error Rate (%) |
|---|---|---|---|---|
| Electromagnetic force (N) | Nonenhanced | 26,962 | 27,116 | 0.57 |
| | Enhanced | 57,918 | 58,474 | 0.95 |
| Inductance gradient ($\mu H \cdot m^{-1}$) | Nonenhanced | 0.6741 | 0.703 | 4.11 |
| | Enhanced | 1.4480 | 1.512 | 4.23 |

According to the data comparison in Table 3, the analytical values of the electromagnetic force and inductance gradient are less than those of the corresponding simulation values. The analytical value error of the nonenhanced inductance was less than that of the enhanced inductance gradient, and the analytical value error rate of the enhanced and nonenhanced inductance gradient was about 4.23% and 4.11%, respectively. The outcome meets the requirements for calculation accuracy.

## 5. Analysis of Influencing Factors of Inductance Gradient

During the process of launching the electromagnetic railgun, the pulse current loaded in the guide rail is characterized by a high peak value, large frequency, and short time. The pulse current produces a strong magnetic field. This strong magnetic field makes the launch environment in the gun chamber extremely complex. The high-speed sliding of the armature in the bore is a transient process of an electromagnetic field, and it is accompanied by changes in the current frequency and armature position. Therefore, it was necessary to analyze the change in inductance gradient with current frequency and armature position to provide theoretical support for subsequent research on the transient inductance gradient of various types of electromagnetic rail launchers. The armature is affected by the current frequency and the armature position, as shown in Table 4.

**Table 4.** Variation in inductance gradient with current frequency and armature position. Unit: ($\mu H \cdot m^{-1}$).

| Structure | Frequency (Hz) | Armature Tail Position (mm) | | | | |
|---|---|---|---|---|---|---|
| | | 30 | 60 | 90 | 120 | 150 |
| Nonenhanced | 100 | 0.8024 | 0.8022 | 0.8002 | 0.7984 | 0.7981 |
| | 1000 | 0.7079 | 0.7065 | 0.7035 | 0.6988 | 0.6865 |
| | 2000 | 0.6783 | 0.6653 | 0.6635 | 0.6613 | 0.6601 |
| | 3000 | 0.6658 | 0.6521 | 0.6485 | 0.6444 | 0.6249 |
| | 5000 | 0.6423 | 0.6396 | 0.6231 | 0.6199 | 0.6101 |
| | 10,000 | 0.6265 | 0.6223 | 0.6208 | 0.6158 | 0.6028 |
| Enhanced | 100 | 1.785 | 1.7833 | 1.7825 | 1.7825 | 1.7825 |
| | 1000 | 1.5201 | 1.5150 | 1.5125 | 1.5100 | 1.5108 |
| | 2000 | 1.4699 | 1.4600 | 1.4596 | 1.4592 | 1.4583 |
| | 3000 | 1.4416 | 1.4392 | 1.4391 | 1.4375 | 1.4367 |
| | 5000 | 1.4285 | 1.4182 | 1.4183 | 1.4175 | 1.4150 |
| | 10,000 | 1.4133 | 1.4000 | 1.3983 | 1.3930 | 1.3917 |

Table 4 shows that as the armature position gradually approaches the muzzle position, the inductance gradient gradually decreases. According to the principle of eddy current field simulation, the armature is stationary at different positions, and the excitation current with the same frequency and amplitude is input, so the current density distribution is

different, thereby decreasing the inductance gradient. From the perspective of the virtual work principle, when the excitation current remains unchanged, the closer the armature is to the muzzle, the more the energy consumption of the launcher increases and the armature kinetic energy decreases, i.e., the electromagnetic thrust received by the armature decreases, thereby decreasing the inductance gradient. Under the same parameters, the gradient of enhanced inductance is about 2.22 times larger than that of nonenhanced inductance.

The inductance gradient of the enhanced four-rail electromagnetic transmitter decreases with the increase in current frequency. This is because the higher the current frequency, the stronger the eddy current effect on the rails and the more obvious the skin effect of the current. That is, as the skin depth becomes shallower, the current flows almost only on the surface of the rail, and the current inside the rail is negligible, thereby decreasing the inductance gradient of the transmitter. The influence of current frequency on the skin depth is shown in Figure 10.

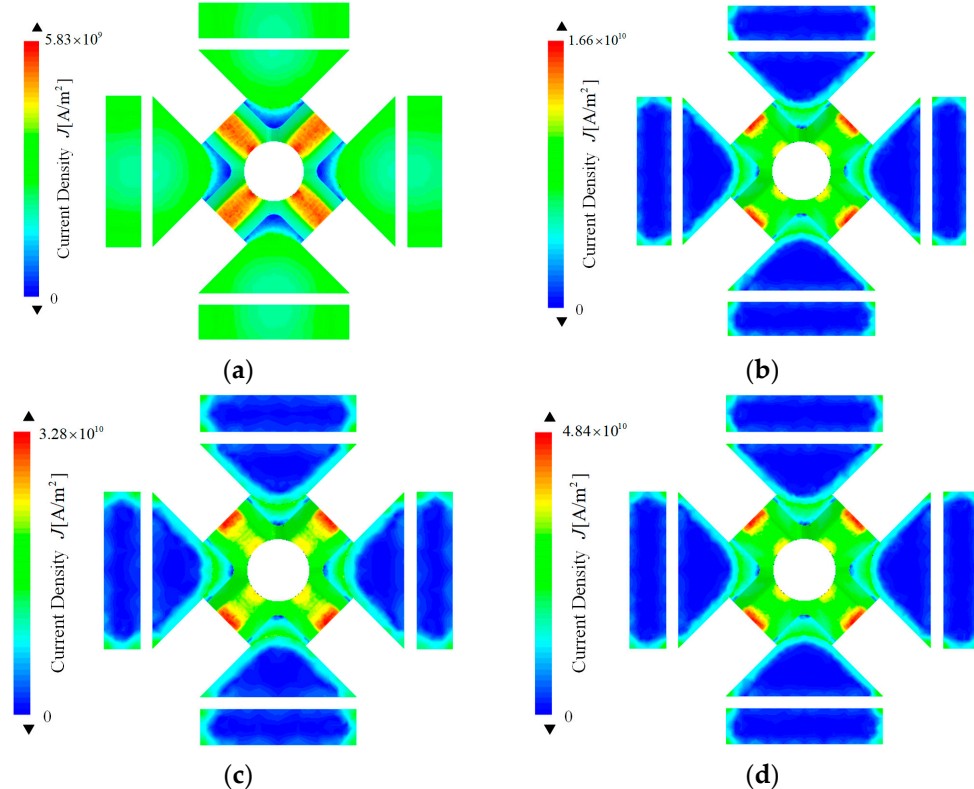

**Figure 10.** Skin depth at different current frequencies: (**a**) 2000; (**b**) 3000; (**c**) 4000; (**d**) 6000 Hz.

## 6. Conclusions

In this study, the electromagnetic characteristics, current skin effect, and proximity effect were fully considered, the line current distribution in the armature–rail structure was simplified, and the simplified model of the current skin effect was obtained. Based on Biot–Savart Law, the analytical formula of the inductance gradient of the model was deduced. This method has a fast calculation speed and produces accurate calculation results. In order to make the analytical calculation model of inductance gradient more realistic to reflect the transient emission process, the calculation error of the analytical method was analyzed with reference to the eddy current field finite element simulation method. Finally, the law of the variation in the inductance gradient with armature position and current frequency was summarized.

(1)    We verified that the error rate of the inductance gradient for enhanced and nonenhanced electromagnetic rail launchers was about 4% using the method proposed in this paper.

(2) Under the same parameters, the gradient of enhanced inductance was about 2.22 times that of nonenhanced inductance. Furthermore, under the same parameters, the inductance gradient decreased with the increase in current frequency; the closer the armature to the muzzle, the smaller the inductance gradient.

(3) In the transient emission process, the current frequency and armature position change with time. When the current frequency remained unchanged, the inductance gradient changed little with the armature position. Therefore, in the transient process, the inductance gradient was mainly affected by the time-varying current frequency.

**Author Contributions:** Conceptualization, G.F.; data curation, S.R. and G.F.; formal analysis, S.R. and X.Z.; investigation, P.Z.; methodology, X.Z.; resources, G.F.; software, S.R. and P.Z.; validation, T.L.; visualization, T.L.; writing—original draft, S.R.; writing—review & editing, G.F. All authors have read and agreed to the published version of the manuscript.

**Funding:** This research received no external funding.

**Data Availability Statement:** The data were generated during the study at Air and Defense College, Air Force Engineering University.

**Conflicts of Interest:** The authors declare no conflict of interest.

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
