# Peer review of "Method of Calculating Inductance Gradient for Complex Electromagnetic Rail Launcher"

_electronics, doi:10.3390/electronics11182912_

Round 1

Reviewer 1 Report

Please read the article again and do some minor changes which are necessary like for example

-when you wrote about reference [9] the author is not the same as the name in the References section

-in some  cases after you wrote the reference in the text there is a point which should not be there(for example [11].); the same when you wrote Figure. 2.

-in Figures 2,3,4 the Legend is not that clear

-Figure 8 should have also a Legend

-Subsection 4 -the title with a big S

-I think the Conclusions need to be rephrased

Reviewer 2 Report

    The authors investigated the electromagnetic and current distribution characteristics of an electromagnetic launcher with a complex structure.

    Based on Biot–Savart Law, an analytical calculation method of inductance gradient for electromagnetic rail launcher with complex structure is proposed. 

Analysis is accomplished in detail.

1. L337-338 : ” ...the analytical value error rate of enhanced and non-enhanced inductance gradient is about 4.23% and 4.11%, respectively.”

; Explain how these calculation errors were estimated. 

2. Figure 10 : Figures are not clear. Use clear figures.
